# DEEP 3D PAN VIA LOCAL ADAPTIVE "T-SHAPED" CONVOLUTIONS WITH GLOBAL AND LOCAL ADAPTIVE DILATIONS

**Juan Luis Gonzalez Bello & Munchurl Kim** *
Department of Electrical Engineering
Korea Advanced Institute of Science and Technology
{juanluisgb,mkimee}@kaist.ac.kr

## ABSTRACT

Recent advances in deep learning have shown promising results in many low-level vision tasks. However, solving the single-image-based view synthesis is still an open problem. In particular, the generation of new images at parallel camera views given a single input image is of great interest, as it enables 3D visualization of the 2D input scenery. We propose a novel network architecture to perform stereoscopic view synthesis at arbitrary camera positions along the X-axis, or "Deep 3D Pan", with "t-shaped" adaptive kernels equipped with globally and locally adaptive dilations. Our proposed network architecture, the monster-net, is devised with a novel t-shaped adaptive kernel with globally and locally adaptive dilation, which can efficiently incorporate global camera shift into and handle local 3D geometries of the target image's pixels for the synthesis of naturally looking 3D panned views when a 2-D input image is given. Extensive experiments were performed on the KITTI, CityScapes, and our VICLAB_STEREO indoors dataset to prove the efficacy of our method. Our monster-net significantly outperforms the state-of-the-art method (SOTA) by a large margin in all metrics of RMSE, PSNR, and SSIM. Our proposed monster-net is capable of reconstructing more reliable image structures in synthesized images with coherent geometry. Moreover, the disparity information that can be extracted from the "t-shaped" kernel is much more reliable than that of the SOTA for the unsupervised monocular depth estimation task, confirming the effectiveness of our method.

## 1 INTRODUCTION

Recent advances in deep learning have pushed forward the state-of-the-art performance for novel view synthesis problems. Novel view synthesis is the task of generating a new view seen from a different camera position, given a single or multiple input images, and finds many applications in robotics, navigation, virtual and augmented reality (VR/AR), cinematography, etc. In particular, the challenging task of generating stereo images given a single input view is of great interest as it enables 3D visualization of the 2D input scene. In addition, the falling price and the increasing availability of the equipment required for VR/AR has fueled the demand for stereoscopic contents.

The previous works, such as Deep3D (Xie et al., 2016), have addressed the right-view generation problem in a fully supervised fashion when the input is the left-view to which the output is the synthetic right-view at a fixed camera shift. In contrast, our proposed Deep 3D Pan pipeline enables the generation of new views at arbitrary camera positions along the horizontal X-axis of an input image with far better quality by incorporating adaptive "t-shaped" convolutions with globally and locally adaptive dilations. Our proposed "t-shaped" kernel with adaptive dilations takes into account the camera shift amount and the local 3D geometries of the target pixels. Panning at arbitrary camera positions allows our proposed model to adjust the baseline (distance between cameras) for different levels of 3D sensation. Additionally, arbitrary panning unlocks the possibility to adjust for different inter-pupillary distances of various persons. Figure 1 shows some generated left and right

---

*https://www.VICLAB.kaist.ac.kr

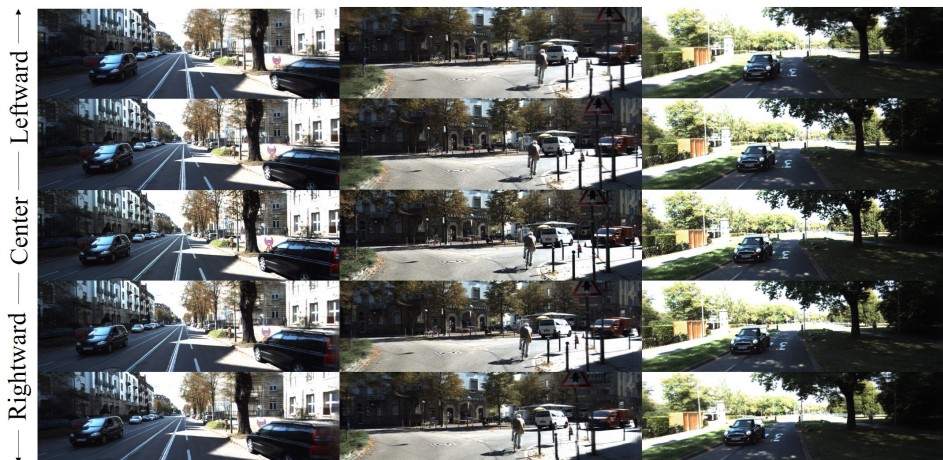

Figure 1: Generated left and right images by our proposed Deep 3D Pan for an input center image.

view images for a given single image input by our proposed Deep 3D Pan pipeline, which we call "monster-net" (**mon**ocular to **ster**eo network). In this paper, we define "panning" in the context of 3D modeling, implying that camera movement is in parallel to the center view camera plane.

In the following sections, we review the related works to stereoscopic view synthesis and discuss the differences with our proposed method, followed by the formulation of our Deep 3d Pan pipeline and finally, we present outstanding results on various challenging stereo datasets, showing superior performance against the previous state-of-the-art methods.

## 2 RELATED WORK

Novel view synthesis is a well-studied problem in deep learning-based computer vision, and has already surpassed the classical techniques for both cases of the multiple-image (Woodford et al., 2007; Liu et al., 2009; Chaurasia et al., 2013) and single-image input (Horry et al., 1997; Hoiem et al., 2005). The latter, single-image based novel view synthesis, is known to be a much more complex problem compared to multiple-image based ones. Previous deep learning-based approaches usually tend to utilize one of the two techniques to generate a novel view: (i) optical flow guided image warping, and (ii) a "flavor" of kernel estimation, also known as adaptive convolutions.

The first technique, **optical flow guided image warping**, has been adopted by several authors to train convolutional neural networks (CNNs) for optical flow or disparity estimation from single or stereo images in an unsupervised fashion. However, their final goal was not to synthesize novel views. These works include those of (Godard et al., 2017; Zhou et al., 2016; Gonzalez & Kim, 2019b; Tosi et al., 2019; Liu et al., 2019; Wang et al., 2019b; Ranjan et al., 2019; Lai et al., 2019). Not all previous existing works have used flow-guided warping for unsupervised training or to regularize supervised methods for optical flow estimation. The work of Im et al. (2019) implemented plane sweep at the feature level to generate a cost volume for multi-view stereo depth estimation. Such plane sweep can be seen as a type of 1D convolution, similar to the 1D kernel utilized in the second approach of kernel estimation for new view synthesis.

On the other hand, the second approach, **kernel estimation or adaptive convolutions**, has proved to be a superior image synthesis technique and has been incorporated in several different ways. For example: (1) Flynn et al. (2016), in their early DeepStereo, formulated a CNN capable of synthesizing a middle view by blending multiple plane-swept lateral view inputs weighted by a "selection volume", which can be interpreted as a 1D (or line-shaped) adaptive convolution; (2) in a similar way, Xie et al. (2016) devised the Deep3D, a non fully-convolutional network that estimates a series of "probabilistic disparity maps" that are then used to blend multiple shifted versions of the left-view input to generate a synthetic right-view image; (3) The adaptive separable convolutions (SepConv) in the work of Niklaus et al. (2017) approximated adaptive 2D convolutions by two (vertical and horizontal) 1D kernels that are applied sequentially to the input current and previous frames ($t_0$ and $t_1$) for the video interpolation problem; (4) In the works of (Zhou et al., 2018; Srinivasan et al.,

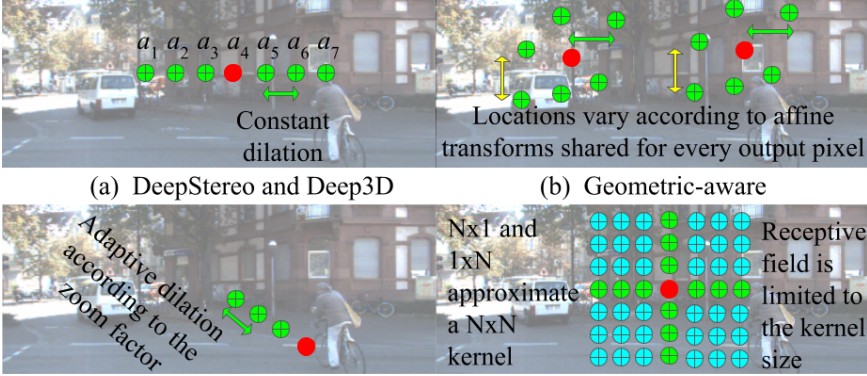

Figure 2: Synthesis techniques based on adaptive convolutions. The background is the input image. Red dots represent target pixel locations in output images. Green (along with red) dots represent sampling positions where the corresponding pixels are used to generate one target pixel.

2019), although with additional considerations, their multiplane image representation approach can be loosely understood as a 1D adaptive convolution as the final operation involves the reduction of a plane sweep volume; (5) Geometric-aware networks in the work of Liu et al. (2018) indirectly achieved adaptive convolutions by learning a fixed number of affine transformations on an input image, where the resulting affine-transformed images are then blended together to generate one output image; and finally, (6) in the work of Gonzalez & Kim (2019a), the authors developed the Deep 3D Zoom Net, which estimates a selection volume for the "blending of multiple upscaled versions of the input image", which can be treated as an special case of a 1D adaptive convolution. The (Flynn et al., 2016) and (Zhou et al., 2018) approaches require two or more images as inputs, thus, greatly reducing the complexity of the synthesis task as most ambiguities are removed by counting on multiple views. In our work, we focus on the single-image based stereoscopic view synthesis task, which is a far more difficult problem as the network needs to understand the 3D geometry in the scene, and to handle complex occlusions, ambiguities and non-Lambertian surfaces.

Although the aforementioned methods are distinguished one another, as the different synthesis techniques have their own properties, they can be all interpreted as belonging to a category of adaptive convolutions which are visualized in Figure 2. As observed in Figure 2-(a), DeepStereo (Flynn et al., 2016) and Deep3D (Xie et al., 2016) share the same shape of kernel, that is, a 1D horizontal-only kernel that samples pixels at a fixed interval, or dilation, along the X-axis for each target output pixel. A 1D horizontal-only constant-dilation kernel suffers from three major drawbacks:

1. Inefficient usage of kernel values. When sampling the positions opposite to the camera movement (which are the pixel locations corresponding to $a_1$-$a_3$ in Figure 2-(a), assuming a rightward camera shift), experiments showed that these kernel values would often be zeros. The same effect repeats when sampling the positions further away from the maximum disparity value of the given scene (which corresponds to the pixel location at $a_7$, assuming that the maximum disparity is 2 and the dilation is 1) as the network is not able to find valid stereo correspondences for these kernel positions;

2. Right-view synthesis is limited to the trained baseline (distance between stereo cameras), as the models over-fit to a specific training dataset with a fixed baseline; and

3. The 1D line kernel has limited occlusion handling capabilities, as the network will try to fill in the gaps with the information contained only along the horizontal direction, limiting the reconstruction performance of the models on the occluded areas.

In contrast, the kernels predicted by the geometric-aware networks (Liu et al., 2018) have deformable structures adaptive to the given input images, as shown in Fig. 2-(b). However, only *one* deformed kernel shape is predicted and shared to synthesize all target output pixels, leading to limited performance. Another drawback of the geometric-aware networks is their complexity, as they require three sub-networks and a super-pixel segmentation step as pre-processing, hindering the processing of high-resolution images. For the Deep 3D Zoom Net (Gonzalez & Kim, 2019a) case (Fig. 2-(c)),

the 1D kernel tends to point to the center of the image, as it performs a blending operation of multiple upscaled versions of the input image. The dilation size of this 1D kernel is adaptive according to the desired 3D-zoom factor. Finally, for the video interpolation case, the SepConv (Niklaus et al., 2017) approximates an NxN adaptive kernel via a 1xN and an Nx1 component (see Fig. 2-(d)) which are sequentially applied to the input images to generate the synthetic output. SepConv has, by design, limited receptive fields, as the dilation size is fixed to 1. Besides, the sequential nature of the kernel forces the vertical component to sample pixels from the output of the horizontal convolution, which could be already degraded due to heavy deformations introduced by the horizontal component.

Recent works have also attempted to improve upon the stereoscopic view synthesis by improving the loss functions involved in the CNN's training. The work of Zhang et al. (2019) proposed a multi-scale adversarial correlation matching (MS-ACM) loss that learns to penalize structures and ignore noise and textures by maximizing and minimizing the correlation-$l_1$ distance in the discriminator's feature-space between the generated right-view and the target-view in an adversarial training setup. Whereas the objective function is a key factor in training any CNN, we believe that, at its current state, the stereoscopic view synthesis problem can benefit more from a better pipeline that can handle the previously mentioned issues and using the widely accepted $l_1$ and perceptual losses (Johnson et al., 2016) for image reconstruction, rather than a more complex loss function.

**Our proposed dilation adaptive "t-shaped" convolutions** incorporate global (new camera position along the X-axis) and local (3D geometries of specific target pixels) information of the input scene into the synthesis of each output pixel value, by not only learning the specific kernel that will generate each output pixel, but also by learning the proper dilation value for each kernel. The "t" shape of the kernel allows the network to account for occlusions by filling-in the gaps (missing information in the output) due to shifted camera positions using not only left-and-right pixels (like DeepStereo and Deep3D), but also up-and-down neighboring pixel information. In addition, the notions of global and local dilations allow our proposed **mon**ocular to **ster**eo network, the monster-net, to generate arbitrarily 3D panned versions of the input center view along the X-axis, a useful feature not present in previous works that allows adjusting for eye-to-eye separation and/or level of 3D sensation.

## 3 METHOD

In order to effectively synthesize an arbitrary 3D panned image, we propose a global dilation filter as shown in Figure 3. Our proposed cross-shaped global dilation filter $\mathbf{T}_d(\boldsymbol{p})$ at a target pixel location $\boldsymbol{p} = (x, y) \in \mathbf{I}_o^t$, where $\mathbf{I}_o^t$ is a generated image, is defined as

$$\mathbf{T}_d(\boldsymbol{p}) = \left[ T_c(x, y), [\mathbf{T}_u, \mathbf{T}_b, \mathbf{T}_l, \mathbf{T}_r]^T \right] \tag{1}$$

where $T_c(x, y)$ is the filter parameter value of $\mathbf{T}_d(\boldsymbol{p})$ at the center location $\boldsymbol{p}$. The upper, bottom, left and right wing parameters ($\mathbf{T}_u, \mathbf{T}_b, \mathbf{T}_l, \mathbf{T}_r$) of the cross-shaped dilation ($d$) filter are defined as

$$
\begin{aligned}
\mathbf{T}_u &= \left[ T_u(x, y - d), T_u(x, y - 2d), \ldots, T_u(x, y - n_u d) \right]^T \\
\mathbf{T}_b &= \left[ T_b(x, y + d), T_b(x, y + 2d), \ldots, T_b(x, y + n_b d) \right]^T \\
\mathbf{T}_l &= \left[ T_l(x - d, y), T_l(x - 2d, y), \ldots, T_l(x - n_l d, y) \right]^T \\
\mathbf{T}_r &= \left[ T_r(x + d, y), T_r(x + 2d, y), \ldots, T_r(x + n_r d, y) \right]^T
\end{aligned}
\tag{2}
$$

where $n_u$, $n_b$, $n_l$ and $n_r$ indicate the numbers of filter parameters in $\mathbf{T}_u, \mathbf{T}_b, \mathbf{T}_l$, and $\mathbf{T}_r$, respectively. For the cross-shaped dilation filter shown in Figure 3, it is more appropriate to have a longer length of the right (left) filter wing than the other three wings when the camera panning is rightward (leftward), as it allows capturing more useful information for the synthesis of a right (left) panned image. In this case, $n_r$ ($n_l$) is set to be greater than $n_l$ ($n_r$), $n_u$ and $n_b$, such that the global dilation filter showed in Figure 3 can be elaborated as a "t-shaped" kernel which can then take into account the camera panning direction for synthesis. Figure 4 shows examples of "t-shaped" kernels overlaid on top of an input center image. As shown in Figure 4-(a), the "t-shaped" kernel has a longer left wing of filter parameters for the synthesis of a leftward camera panning while in Figure 4-(b) it shows a longer right-wing of filter parameters for the synthesis of a rightward camera panning.

**Why "t" shape?** Experiments with symmetric kernel shapes (e.g., "+" shape) were performed first, but it was noted that most of the elements on the left (right), upper and bottom sides against the

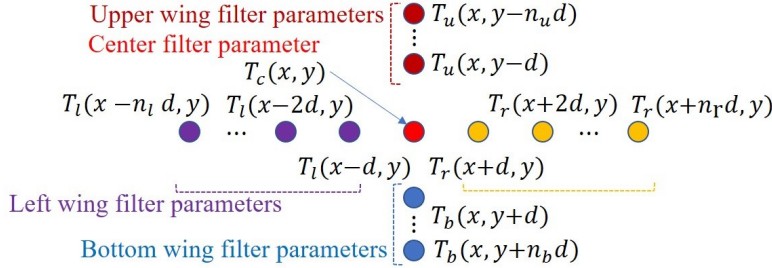

Figure 3: Our proposed global dilation ($d$) filter with a general cross shape.

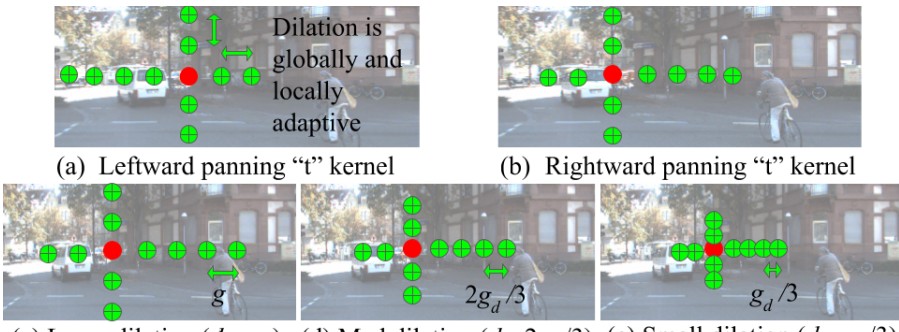

(a) Leftward panning "t" kernel    (b) Rightward panning "t" kernel

(c) Large dilation ($d_1{=}g_d$)  (d) Med dilation ($d_2{=}2g_d/3$)  (e) Small dilation ($d_3{=}g_d/3$)

Figure 4: Our proposed "t-shaped" kernels are overlaid on top of a center input image. The distance between samples (dilation) is adaptive according to the amount and direction of 3D panning to be applied to the input image and the local 3D geometry of the scene.

centered red dot of the kernel tended to have very small values close to zeros for most target pixels for the rightward (leftward) movement of the camera. Similar to SepConv (Niklaus et al., 2017), the experiments with a horizontal kernel applied first followed by a vertical kernel were performed, yielding poor results. It was discovered that the **"t" shaped kernel is more efficient than the "+" shaped kernel** as it picks up more effective sampling positions with a fewer parameters than the standard adaptive convolutions such as those in SepConv. As depicted in Figure 5, the "t-shaped" kernels can embed useful information like disparity and occlusion from a monocular image into the stereo synthesis process.

**The longer right (left) wing of the "t-shaped" kernel contains disparity information**, as it will try to sample pixels from the right (left) side to the target pixel when the camera is assumed to move in the rightward (leftward) direction. Figure 5-(a) depicts a primitive disparity map $\mathbf{D}_p$ that was constructed by the weighted sum of the kernel values in the longer kernel wing, as described by

$$\mathbf{D}_p(\boldsymbol{p}) = \sum_{i=1}^{n_r} \frac{i}{n_r} T_r(x + id, y) \tag{3}$$

where $T_r(x + id, y)$ is the $i$-th value of the longer wing $\mathbf{T}_r$ at pixel location $\boldsymbol{p} = (x, y)$ for the rightward 3D panning of an input center image $\mathbf{I}_c$. Note that $\mathbf{D}_p$ is normalized in the range $[0, 1]$. Interestingly, as shown in Figure 5-(a), the generated disparity map looks very natural and appropriate, which implies the effectiveness of our "t-shaped" kernel approach.

**The short left (right), upper and bottom wings of the "t-shaped" kernel contain occlusion information**, as the network will try to fill in the gaps utilizing surrounding information that is not present in the long part of the "t-shaped" kernel. It is also interesting to see the occlusion map in Figure 5-(b) where a primitive rightward occlusion map $\mathbf{O}_p^r$ was constructed by summing up the "t-shaped" kernel values in the short wing parts according to the following:

$$\mathbf{O}_p^r(\boldsymbol{p}) = \sum_{i=1}^{n_l} T_l(x - id, y) + \sum_{i=1}^{n_u} T_u(x, y - id) + \sum_{i=1}^{n_b} T_b(x, y + id) \tag{4}$$

The bright regions or spots in Figure 5-(b) indicate the occlusions due to the camera shift along the horizontal axis of the input center image, which are likely to happen for the case of the camera's

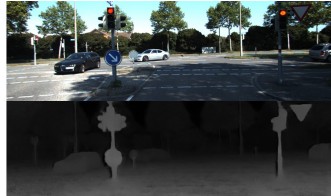 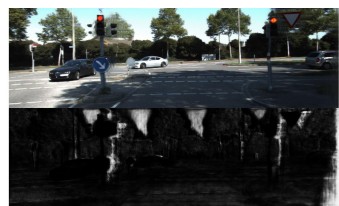

(a) Input center image and disparity map    (b)  Output panned image and occlusion map

Figure 5: Disparity ($\mathbf{D}_p$) and occlusion ($\mathbf{O}_p^r$) maps generated from the proposed "t-shaped" kernel.

rightward panning. For both Equations (3) and (4), the primitive disparity and occlusion maps for the leftward panning case can be obtained by swapping the $r$ and $l$ indices.

### 3.1    GLOBALLY AND LOCALLY ADAPTIVE DILATIONS FOR THE SYNTHESIS OF A NEW VIEW IMAGE AT A SHIFTED CAMERA POSITION

In general, the disparity amounts between stereo images are variable at different pixel locations according to the distance between stereo cameras and the local scene geometries. Therefore, it is necessary to take into account the variable disparity in synthesizing a new view in a globally and locally adaptive fashion. For this, a "t-shaped" kernel is introduced with a controllable dilation factor by which both camera shift and local changes in image geometry can be effectively taken into account when synthesizing a new (left or right) view for the input center image. Any kernel with a fixed dilation may cause a limited accuracy in synthesizing a novel view because the disparity amounts vary over the whole image according to the cameras' baseline and the local geometries. So, our "t-shaped" kernel is proposed to make the synthesis of novel views not only globally, but locally adaptive to the camera shift and its local changes in image geometry by controlling its dilation size per-pixel in the output image. Globally, a short dilation value is more appropriate when slightly shifting the camera, while a high dilation value is desirable when largely shifting the camera position. In a local manner, a small dilation value is appropriate for far-away objects from the camera while very close objects to the camera can be better reconstructed with a larger dilation value.

#### 3.1.1    GLOBAL DILATION

We define the global dilation $g_d$ as the pixel distance between two consecutive kernel sampling positions, which is given by the pan amount $P_a$ to be applied to the input center image $\mathbf{I}_c$ divided by the total number of filter parameters in the longer "t-shaped" kernel wing ($n_l$ or $n_r$). $P_a$ has a unit of pixels mapped in the image corresponding to the camera shift into the left or right direction and takes on floating numbers. Therefore, the global dilation $g_d$ is given by

$$g_d = \{\ P_a/n_r \quad if\ P_a > 0, \quad P_a/n_l \quad if\ P_a < 0\ \} \tag{5}$$

where $P_a$ takes on positive (negative) values for the rightward (leftward) panning scenario. The pan amount needed to generate a left-view or a right-view is determined during training according to the closest possible objects to the camera. The "closest possible objects" vary over different training datasets. For our novel view synthesis task, like in (Godard et al., 2017; Gonzalez & Kim, 2019b), we assume the KITTI dataset to have a maximum or "closest possible object" disparity of 153 pixels. During training, $P_a$ is set to 153 and -153 for the rightward and leftward panning, respectively.

#### 3.1.2    LOCAL DILATION

While global dilation allows the "t-shaped" kernel to take into account the global camera shift, a locally adaptive mechanism is needed to synthesize new views of locally variable disparity. Such a mechanism is realized by first generating multiple images with the "t-shaped" kernel at $N$ different dilations and blending them per-pixel in a locally adaptive manner. The blending is a weighted sum of filtered images by the "t-shaped" kernel with $N$ different dilations, where the blending weights $(w_1, w_2, \ldots, w_N)$ control the local dilation per-pixel and are learned via a convolutional neural network (CNN) along with the parameter values of the "t-shaped" kernel. Let $|g_d|$ be the maximum dilation value that is a fractional number. Figures 4-(c), -(d) and -(e) illustrative three "t-shaped" kernels with a maximum dilation $|g_d|$ and two dilation values less than $|g_d|$. To generate an output

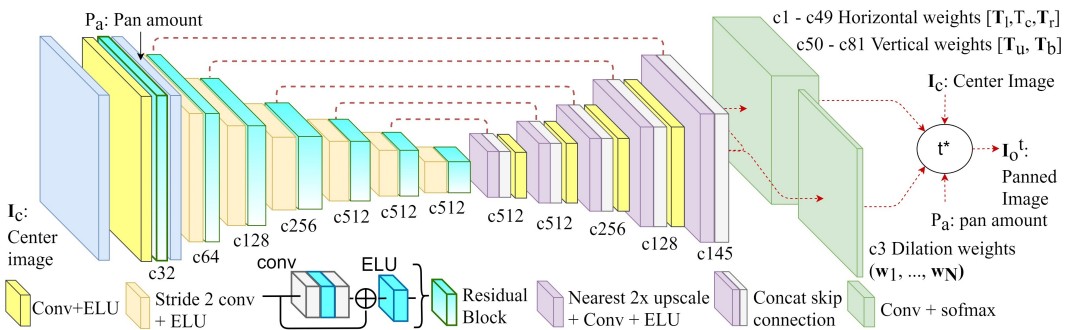

Figure 6: Our t-net architecture. The t-net estimates the kernel values and the dilation weights used for the local adaptive t convolutions with global and local adaptive dilation.

image $\mathbf{l}_o^t$ panned to the rightward direction ($g_d > 0$) or to the leftward direction ($g_d < 0$), the input center image $\mathbf{l}_c$ is first filtered by $N$ "t-shaped" kernels $\mathbf{T}_{d_i}$ of different dilations ($d_1, \ldots, d_N$). Then, local adaptive dilations are calculated by linearly combining the resulting $N$ intermediate filtered images according to the corresponding blending weights ($w_1, w_2, \ldots, w_N$). Based on the $N$ different global dilations, the output image value $I_o^t(\boldsymbol{p})$ at a pixel location $\boldsymbol{p}$ can be calculated as

$$I_o^t(\boldsymbol{p}) = \sum_{i=1}^N w_i(\boldsymbol{p}) \left[ \mathbf{l}_c * \mathbf{T}_{d_i} \right] (\boldsymbol{p}) \tag{6}$$

where $[\mathbf{l}_c * \mathbf{T}_{d_i}](\boldsymbol{p})$ is a "t-shaped" convolution at location $\boldsymbol{p}$ between $\mathbf{l}_c$ and $\mathbf{T}_{d_i}$ of dilation $d_i = (1 + (1 - i)/N)g_d$ for $i = 1, \ldots, N$. $w_i(\boldsymbol{p})$ indicates a blending weight for the $i$-th global dilation.

## 3.2 NETWORK ARCHITECTURE

We propose an end-to-end trainable CNN, called the "monster-net" (**mon**ocular to **ster**eo net). The monster-net is made of two main building blocks, a novel view synthesis network, the "t-net", and a resolution restoration block, the "sr-block". Given an input center image $\mathbf{l}_c$ and pan amount $P_a$, the final output panned image $\mathbf{l}_o$ is obtained by sequentially applying the aforementioned modules by

$$\mathbf{l}_o = monster\text{-}net(\mathbf{l}_c, P_a) = sr\text{-}block(t\text{-}net(\mathbf{l}_c, P_a; \theta_t), \{\mathbf{l}_{cs}^n\}; \theta_{sr}) \tag{7}$$

where $\theta_t$ and $\theta_{sr}$ parameterize the t-net and the sr-block respectively. $\{\mathbf{l}_{cs}^n\}$ is the stack of progressively shifted-downscaled versions of the input center image $\mathbf{l}_c$ described in the SR-BLOCK section.

**The t-net**. The "t-net" estimates both the "t-shaped" global dilation kernel parameters ($\mathbf{T}_d$) and the adaptive local dilation weights ($w_1, w_2, \ldots, w_N$). The t-net is designed to have large receptive fields to synthesize detailed image structures of a new view image, which corresponds to a shifted camera position. Such large receptive fields are useful in capturing the global image structure and contextual information needed for a new view image to be synthesized. For this, an auto-encoder with skip connections (not a U-net structure) is adopted, which allows the t-net to have effectively large receptive fields and to efficiently fuse global and local (fine details) information on the decoder stage. For better feature extraction, we adopt the residual connections in the encoder side, as proposed by (Gonzalez & Kim, 2019b). The t-net estimates all necessary values to perform the operation described by Equation (6). The t-net, depicted in Figure 6, has two output branches: the first output branch yields 81 channels, where the first 49 are horizontal parameter maps, and the following 32 vertical parameter maps; the second output branch generates the 3-channel blending weight maps for the local adaptive dilation. That is, each channel-wise vector at a pixel location for the first output branch corresponds to the t-kernel parameter values $[T_c, \mathbf{T}_l^T, \mathbf{T}_r^T, \mathbf{T}_u^T, \mathbf{T}_b^T]$, and each channel-wise vector for the second output branch corresponds to the blending weights $[w_1, w_2, \ldots, w_N]$, utilized for local dilations in Equation (6). As our t-net is devised to generate arbitrarily panned novel views, feeding the pan amount as a 1-channel constant feature map ($P_a(\boldsymbol{p}) = P_a \forall \boldsymbol{p}$) helps the network take into account the varying pan direction, and the amount of occlusion on the 3D panned output. The effect of feeding the pan amount is further discussed in appendix A-1.

**Super resolution (SR) block.** As generating a full resolution dilation-adaptive t-kernel would be computationally too expensive, we propose to estimate it at a low resolution (LR) for the synthesis

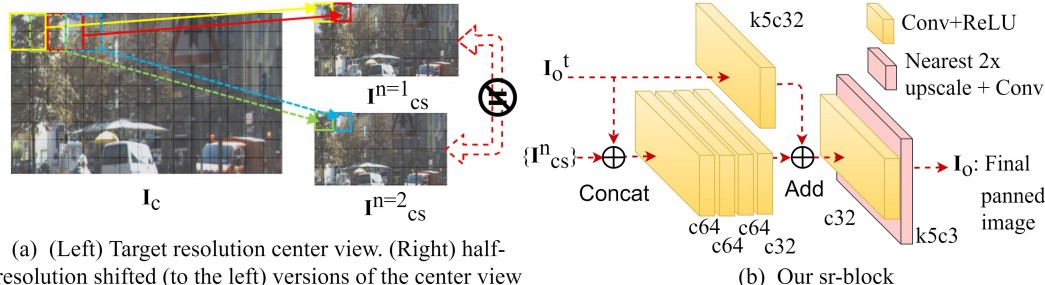

(a) (Left) Target resolution center view. (Right) half-resolution shifted (to the left) versions of the center view

(b) Our sr-block

Figure 7: (a) Shifted-LR versions of the center-view contain different information as they are sampled from different groups of pixels via bilinear interpolation depending on the stride (controlled by the maximum disparity). (b) Our light sr-block. All convs have 3x3 kernels otherwise specified.

Table 1: Stereoscopic view synthesis performance on the 400 KITTI2015 training images (left) and the 500 CityScapes validation images (right). $l_p$: perceptual loss. ↑↓ indicate the better performance.

| Model | training dataset | loss | RMSE↓ | PSNR↑ | SSIM↑ | RMSE↓ | PSNR↑ | SSIM↑ |
|---|---|---|---|---|---|---|---|---|
| Deep3D | K | $l_1$ | 26.13 | 20.07 | 0.637 | 30.10 | 18.82 | 0.655 |
| Deep3D-B | K | $l_1 + l_p$ | 26.00 | 20.10 | 0.633 | 31.34 | 18.46 | 0.636 |
| SepConv | K | $l_1$ | 27.22 | 19.73 | 0.633 | 27.77 | 19.54 | 0.660 |
| SepConv-D | K | $l_1 + l_p$ | 26.36 | 20.02 | 0.626 | 29.66 | 18.95 | 0.647 |
| monstet-net | K | $l_1 + l_p$ | **25.61** | **20.24** | **0.641** | 20.28 | 22.34 | 0.710 |
| monster-net | K+CS | $l_1$ | **24.11** | **20.76** | **0.667** | **12.87** | **26.36** | **0.816** |
| monster-net (full) | K+CS | $l_1 + l_p$ | 24.44 | 20.64 | 0.651 | 13.12 | 26.20 | 0.805 |
| monster-net | K+CS+VL | $l_1 + l_p$ | 24.62 | 20.55 | 0.645 | - | - | - |

of a half-resolution novel view, and then to apply deep learning based SR techniques to bring the LR novel view to the high (or original) resolution (HR). In comparison, in Deep3D and SepConv, the estimated LR kernel is upscaled with conventional methods to the HR and then applied to the input image(s), which is a costly operation as it is carried out in the HR dimensions and can lead to blurred areas as the kernel is just bilinearly interpolated. In our proposed pipeline, instead of utilizing common single image SR methods like (Dong et al., 2015; Shi et al., 2016; Kim et al., 2016), we propose to apply a stereo-SR method. The stereo-SR technique in (Jeon et al., 2018) takes a LR stereo pair (left and right views) as input and progressively shifts the right-view producing a stack that is concatenated with the left-view and later processed by a CNN to obtain the super-resolved left-view. This process is made at an arbitrary and fixed stride (e.g. 1 pixel at every step of the stack) and does not take into account the maximum disparity between the input views. For our Deep 3D Pan pipeline, we propose to use the maximum disparity prior that can be obtained from the long wing of the t-kernel to dynamically set the shifting stride. Additionally, instead of interpolating and processing the low resolution panned view $I_o^t(\boldsymbol{p})$ on the HR dimensions, we progressively shift and then downscale the high-resolution center view $\mathbf{I}_c$ by a factor of x2. This allows our sr-block to operate on the LR dimensions without performance degradation, as high frequency information in the horizontal axis is not lost but distributed along the levels of the shifted center view stack as depicted in Figure 7-(a). Our sr-block, depicted in Figure 7-(b), is a simple, yet effective module that takes as input the LR $\mathbf{I}_o^t$ view and the shifted-downscaled center view stack $\mathbf{I}_{cs}^n$ described by

$$\mathbf{I}_{cs}^n = g(\mathbf{I}_c, \frac{nP_a}{N_s} max(\mathbf{D}_p)) \tag{8}$$

where $g(\mathbf{I}, s)$ is an $s$-strided horizontal-shift and 2x down-scaling operator applied on image $\mathbf{I}$. The stride $s$ can take any real number and the resulting image is obtained via bilinear interpolation. $N_s$ is the depth of the stack, and was set to $N_s = 32$ for all our experiments). The stack is concatenated with the LR $\mathbf{I}_o^t$ and passed trough four Conv-ReLU layers followed by a residual connection as shown in Figure 7-(b). The final step up-scales the resulting features into the target resolution via nearest interpolation followed by a convolutional layer. The last layer reduces the number of channels to three for the final RGB output $\mathbf{I}_o$. Nearest upscaling was adopted as it yields no checkerboard artifacts in contrast with transposed or sub-pixel convolutions (Niklaus et al., 2017).

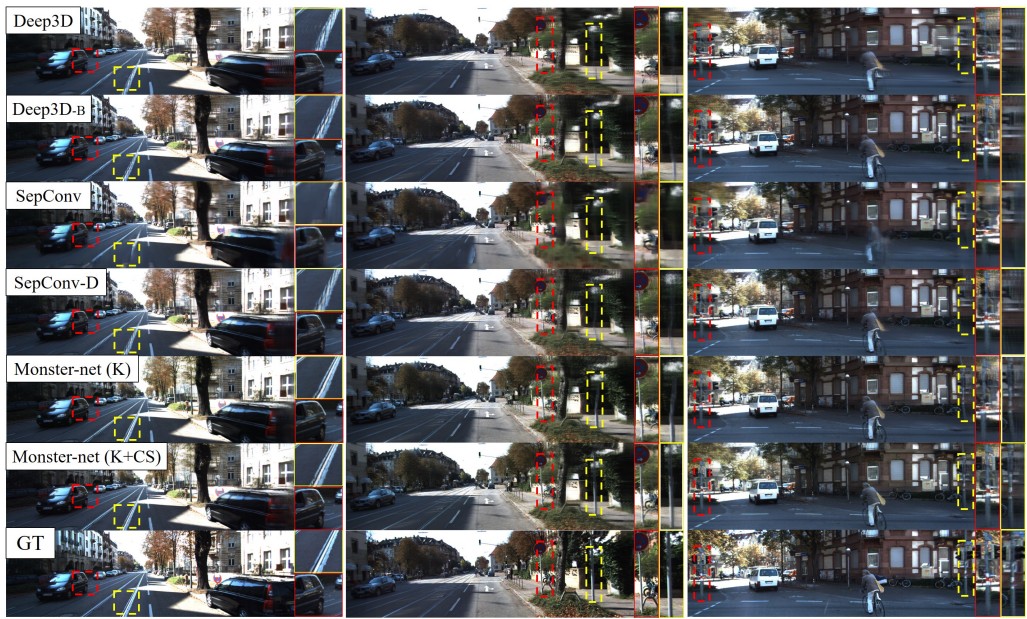

Figure 8: Comparison against the state-of-the-art methods for stereoscopic view synthesis.

# 4 EXPERIMENTS AND RESULTS

To demonstrate the effectiveness of our "t-shaped"-dilation-adaptive kernel, we performed several experiments on the challenging KITTI2012 (Geiger et al., 2012), KITTI2015 (Menze & Geiger, 2015), and CityScapes (Cordts et al., 2016) datasets. As these stereo datasets only consist of outdoor scenes, we also performed experiments on our indoors dataset, called the VICLAB_STEREO dataset. Surprisingly, to our best knowledge, this is the first stereo dataset available that focuses on the indoor scene, which is planned to be publicly available for research. Additionally, our formulation of global and local adaptive dilations allows our monster-net to be trained on multiple stereo datasets at the same time, even if these have different baselines. Instead of over-fitting on a single camera baseline like the previous methods (Xie et al. (2016); Godard et al. (2017); Zhang et al. (2019); Luo et al. (2018)), our monster-net can build knowledge when simultaneously trained on many datasets. To our best knowledge, our Deep 3D Pan pipeline is the first method designed to be trained on multiple baseline datasets concurrently for the **stereoscopic view synthesis** problem where **unsupervised monocular depth estimation** is even used particularly. For more details about the datasets and multi-dataset training, please see the appendix A-3.

We compare our monster-net against the stereoscopic view synthesis SOTA: Deep3D Xie et al. (2016) and a version of SepConv (Niklaus et al., 2017) modified for right-view synthesis. Firstly, for a fair comparison, the backbone convolutional auto-encoders for the Deep3D and SepConv were set up to be equivalent to our t-net's, that is, a six-stage encoder-decoder with skip connections and residual blocks in the encoder side. Secondly, we compare our monster-net with Deep3D-B, a "Bigger" version of Deep3D, where, instead of 32 elements in the 1D kernel as in its original work, we use 49 elements to match the number of horizontal kernel values in our t-net. Thirdly, we compare against SepConv-D, a dilated version of the SepConv such that the receptive field of the separable convolutions has a size of 153x153. The Deep3D and the SepConv models are trained without using perceptual loss as in their original works. For a more meaningful comparison, the Deep3D-B and the SepConv-D are trained with a combination of $l_1$ and perceptual loss $l_p$ (Johnson et al., 2016), and demonstrate that a better loss function than $l_1$ does not contribute enough to the stereoscopic view synthesis problem. For more implementation details, reefer to the appendix A-4.

Additionally, we compare the quality of the embedded disparity in the long wing of the "t-shaped" kernel with those of the state-of-the-art models for the monocular depth estimation task. For that, we first define a disparity refinement sub-network that uses the primitive disparity obtained from the long wing of the "t-shaped" kernel as prior information. Secondly, we define a special post-processing (spp) step, which, instead of relying on a naive element wise summation as in Godard

Table 2: Depth metrics (Eigen et al., 2014) for KITTI2015. Models are trained with video (V), stereo (S), semi global matching (SMG) or GT depth (Supp). Top models in terms of $a^1$ accuracy are highlighted. Simplified table, see appendix A.9 for the full version.

| Model | Supp | V | S | dataset | abs rel↓ | sq rel↓ | rms↓ | log rms↓ | $a^1$ ↑ | $a^2$ ↑ | $a^3$ ↑ |
|---|---|---|---|---|---|---|---|---|---|---|---|
| Wang et al. (2019a) (9-view) | | x | | K | 0.112 | 0.418 | 2.320 | 0.153 | 0.882 | 0.974 | 0.992 |
| Tosi et al. (2019) (pp) | SMG | | x | K+CS | 0.096 | 0.673 | 4.351 | 0.184 | 0.890 | 0.961 | 0.981 |
| **ours with refine block (spp)** | | | x | K+CS | 0.099 | 0.950 | 4.739 | 0.160 | **0.900** | 0.971 | 0.989 |
| Gur & Wolf (2019) | x | | | K | 0.110 | 0.666 | 4.186 | 0.168 | 0.880 | 0.966 | 0.988 |
| Luo et al. (2018) | x | | | K | 0.094 | 0.626 | 4.252 | 0.177 | 0.891 | 0.965 | 0.984 |
| **Wang et al. (2019a) (1/9-view)** | x | x | | K | 0.088 | 0.245 | 1.949 | 0.127 | **0.915** | 0.984 | 0.996 |

et al. (2017), takes into account the ambiguities of the first and second forward passes to generate a remarkable sharp and consistent disparity map. For more details on the refinement block and our special post-processing, reefer to the appendix A-2.

## 4.1 RESULTS ON THE KITTI, CITYSCAPES AND THE VICLAB_STEREO DATASETS

Table 1 shows the performance comparison for our method and previous works. It is important to mention that our monster-net performs inference on full resolution images, while the previous approaches for single-view novel view synthesis perform estimation on reduced resolution inputs. Our method outperforms the Deep3D baseline by a considerable margin of 0.7dB in PSNR, 2.0 in RMSE, and 0.03 in SSIM. The qualitative results are shown in Figure 8. Our method produces superior looking images. In Deep3D and SepConv, many objects appear too blurred such that their boundaries can hardly be recognized in the synthetic images (e.g the motorcycle, persons, traffic signs, etc.). We challenged the models trained on KITTI (K) to perform inference on the CityScapes validation split (CS), and observed that our method generalizes much better than the Deep3D baseline with up to 3dB higher in PSNR. When training the monster-net with K+CS, we get an additional improvement of 4dB PSNR in the validation CS dataset. Incorporating an indoor dataset to our training pipeline is also possible, making our network applicable to a wide variety of scenarios. We added the VI-CLAB_STEREO (VL) dataset to the training, that is K+CS+VL, and observed little impact on the K dataset performance as shown in Table 1. We also tested the performance of our monster-net on the validation split of the VL dataset. We observed that our full monster-net trained on K+CS achieved a mean PSNR of 19.92dB, while achieving a mean PSNR of 21.78 dB when trained on K+CS+VL. For a network trained on the outdoors dataset only it is difficult to generalize to the indoors case, as the latter contains mainly homogeneous areas, whereas the outdoors case mainly contains texture rich scenes. Visualizations on CS and VL, and ablation studies that prove the efficacy of each of our design choices can be found in the appendices A-5, A-6 and A-8.

## 4.2 RESULTS ON DISPARITY ESTIMATION

With the addition of a relatively shallow disparity refinement sub-network, the monster-net remarkably outperforms all the state-of-the-art models for the unsupervised monocular depth estimation task, as shown in Table 2. Our monster-net with disparity refinement even outperforms the supervised monocular disparity estimation methods such as (Luo et al., 2018; Gur & Wolf, 2019) and multiple view unsupervised methods such as (Wang et al., 2019a; Ranjan et al., 2019).

## 5 CONCLUSION

We presented an adaptive "t-shaped" kernel equipped with globally and locally adaptive dilations for the Deep 3D Pan problem, defined as the task of arbitrarily shifting the camera position along the X-axis for stereoscopic view synthesis. Our proposed monster-net showed superior performance to the SOTA for right-view generation on the KITTI and the CityScapes datasets. Our monster-net also showed very good generalization capabilities with 3dB gain in PSNR against the Deep3D baseline. In addition, our method presents no-discontinuities, consistent geometries, good contrast, and naturally looking left or right synthetic panned images. Our monster-net can be extended for image registration, monocular video to stereo video, and generation of novel views at any camera translation by just allowing a pixel-wise rotation of our "t-shaped" kernel.

ACKNOWLEDGMENTS

This work was supported by Institute for Information & communications Technology Promotion (IITP) grant funded by the Korea government (MSIT) (No. 2017-0-00419, Intelligent High Realistic Visual Processing for Smart Broadcasting Media).

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

## A  APPENDIX

### A.1  EFFECT OF FEEDING THE PAN AMOUNT TO THE T-NET

The larger the pan amount, the greater the occlusions to be handled in the synthetic output image. The effect of feeding the pan amount $P_a$ as a one-channel constant feature to the t-net can be visualized in Figure 9, where multiple primitive disparity and occlusion maps are depicted for different pan amounts. As shown in Figure 9, the network generates different maps for different magnitudes and directions of $P_a$ while keeping the input center image $I_c$ unchanged, confirming the effect of the pan amount as prior knowledge to the network. The difference between the disparity maps can be appreciated in the "holes" or "shadows" casted in the objects borders, as they represent the occluded content seen from the output 3D panned image. In the red box in Figure 9 it is observed that the shadows casted by leftward and rightward camera panning appear in opposite sides ob the objects. In the yellow box, it is observed that the larger the $P_a$, the larger the shadows projected, as more occlusions are to be handled.

### A.2  DISPARITY REFINEMENT SUB-NETWORK

The disparity refinement network architecture, as depicted in Figure 10, has two input branches: one takes a stack of the 2x bilinearly upscaled center image disparity prior ($\mathbf{D}_{cp}$) and the RGB center image ($\mathbf{I}_c$); and the other is fed with a stack of the 2x bilinearly upscaled output panned view disparity prior ($\mathbf{D}_{op}$) and the generated panned view ($\mathbf{I}_o$). The disparity refinement block is a relatively shallow auto-encoder with skip connections and rectangular convolutions as fusion layers in the decoder stage. This allows to increase the receptive field size in the horizontal axis, thus improving the stereo matching performance, as suggested by (Gonzalez & Kim, 2019b). We configure the output layer of our refinement network with the last layer of Gonzalez & Kim (2019b)'s architecture, which allows to do ambiguity learning in our refinement block. Ambiguity learning allows the network to

unsupervisedly account for occlusions and complex or clutered regions that are difficult to minimize in the photometric reconstruction loss (Gonzalez & Kim, 2019b). We train the refinement block with the loss functions defined in (Gonzalez & Kim, 2019b) and a new additional loss towards producing the refined disparity maps similar to the primitive disparity maps $\mathbf{D}_{cp}$ and $\mathbf{D}_{op}$. The refinement network is encouraged to produce refined center and panned disparity maps $\mathbf{D}_c$ and $\mathbf{D}_o$ similar to their primitive counterparts of half the resolution (as they are estimated from the t-net), by minimizing the following primitive disparity loss

$$l_{Dp} = ||\mathbf{D}_{op} - \mathbf{D}_o^{1/2}||_1 + ||\mathbf{D}_{cp} - \mathbf{D}_c^{1/2}||_1 \qquad (9)$$

where $\mathbf{D}_c^{1/2}$ and $\mathbf{D}_o^{1/2}$ are the bilinearly downscaled and refined center and panned disparity maps by a factor of 1/2, respectively. We give a weight of 0.5 to this new loss term. The disparity refinement block can be trained end-to-end along with the monster-net or from a pre-trained monster-net.

### A.2.1 Special post-processing

Instead of relying on naive post-processing approaches like in (Godard et al., 2017), which consist on running the disparity estimation twice with normal and horizontally flipped inputs and then taking the average depth, we define a novel special post-processing step (spp) by taking into account the ambiguities in the first and second forward pass. We noted that the ambiguity learning from (Gonzalez & Kim, 2019b), which we incorporate in our disparity refinement block, can be used to blend the resulting disparities from the first and the second forward pass such that only the best disparity estimation (or ambiguity free) from each forward pass is kept on the final post-processed disparity. Figure 11 depicts our novel post-processing step, which consist on running the forward pass of our monster-net with disparity refinement block with $P_a = 153$ and $P_a = -153$, for the first and the second pass respectively. Then, the generated ambiguity masks of each forward pass are concatenated to create a two-channel tensor and passed through a softmax operation along the channel axis. The resulting soft-maxed ambiguities are used to linearly combine the disparity maps of each forward pass. As can be observed in Figure 11, the soft-maxed ambiguity mask effectively segment the best disparity estimation from each forward pass. Figure 12 shows the primitive disparity map, the subsequent refinement step, the naive post-processing (pp) and our novel post-processing (spp).

### A.3 The KITTI, CityScapes and VICLAB_STEREO datasets

The KITTI dataset is a very large collection of mid-resolution 370x1226 stereo images taken from a driving perspective. We used the KITTI split as suggested in (Godard et al., 2017), which consists of 29,000 stereo pairs from 33 different scenes of the KITTI2012 dataset. We set apart the KITTI2015 dataset for validation as it contains 400 images excluded from the KITTI split. Additionally, the KITTI2015 contains sparse disparity ground truths (GTs) which are obtained from LIDAR and then are refined by car CAD models. We use these GTs to evaluate the quality of the estimated disparity that can be extracted from the long wing of the t-kernel. CityScapes is a higher resolution stereo dataset with 24,500 stereo pairs that we extract from the $train$, $test$ and $train\_extra$ directories for training. The $val$ directory is left for validation with 500 stereo pairs. We pre-process the CityScapes dataset for faster and more robust training. We first remove the top 25, bottom 200

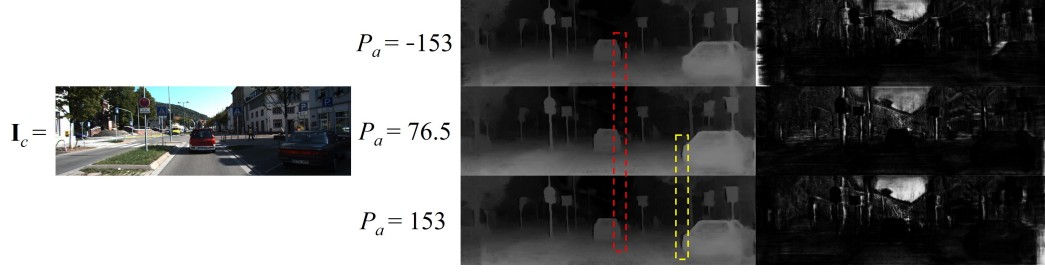

Figure 9: Effect of feeding different values of $P_a$ while keeping the input image unchanged. Different values of $P_a$ generate different occlusions and different holes in disparities (see red and yellow boxes), which indicate the occluded regions in the target panned image.

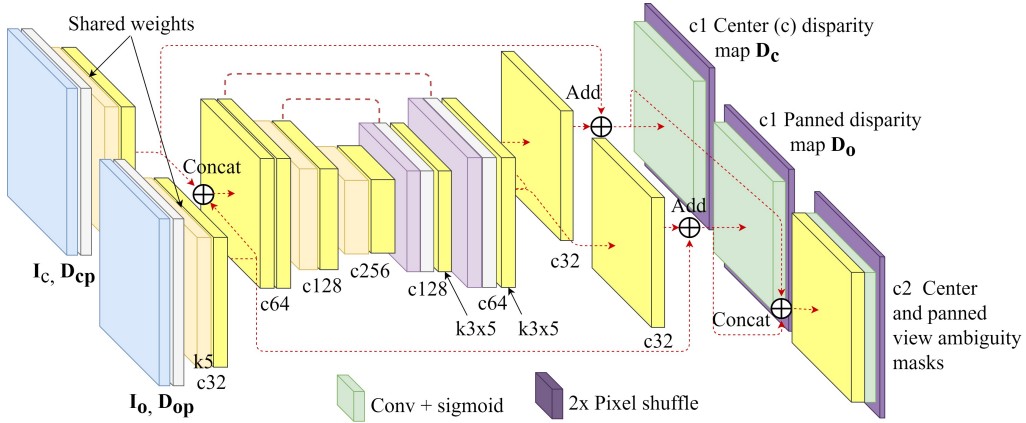

Figure 10: Disparity refinement block

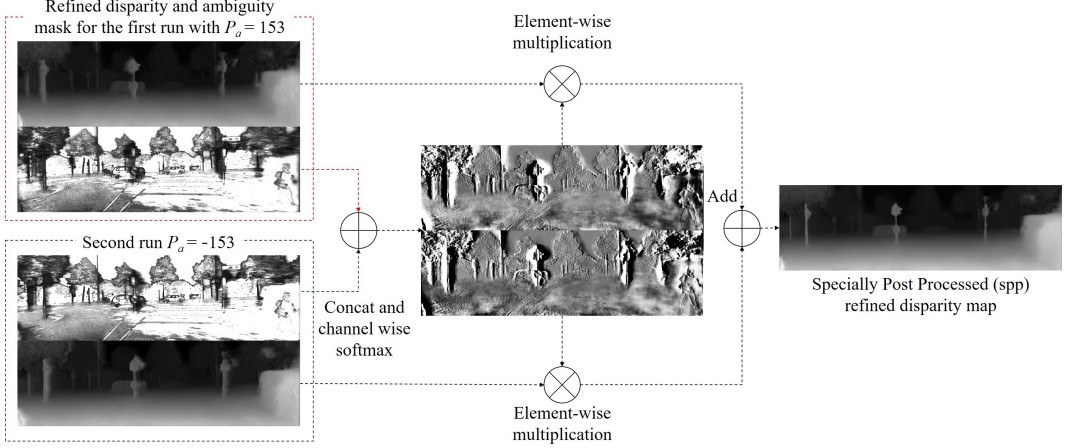

Figure 11: Our novel special post-processing step (spp)

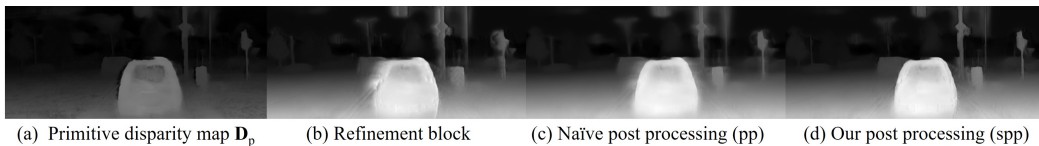

(a) Primitive disparity map $\mathbf{D}_p$    (b) Refinement block    (c) Naïve post processing (pp)    (d) Our post processing (spp)

Figure 12: Primitive disparity and different refinement options.

and left 100 pixels to avoid car hoods and rectification artifacts, thus yielding a final image size of 799x1948. Secondly, we save the cropped images in $.jpg$ format to accelerate loading times during training. The VICLAB_STEREO is an indoor left and right view dataset that was captured in 18 different buildings on a static pedestal using a Stereolabs's ZED$^{\text{TM}}$stereo camera. The captured 2,051 high-resolution stereo pairs were rectified via checkerboard calibration as described in this MATLAB website[1]. After rectification, the images in the VICLAB_STEREO yield a final resolution of 1247x2454. We use 2,035 stereo pairs spanning 17 buildings for training, and the remaining 16 images for validation.

### A.3.1 TRAINING ON MULTIPLE DATASETS

To our best knowledge, our Deep 3D Pan pipeline is the first method designed to be trained on multiple baseline datasets at the same time for the **stereoscopic view synthesis** problem and the **unsupervised monocular depth estimation** task. It should be noted that the work of Facil et al. (2019) only handled the supervised monocular depth estimation task for multiple datasets with different camera intrinsics utilizing "CAM-Convs", which is a simpler problem than our unsupervised problem. While they require to know the intrinsic matrix for each dataset, along with added computational complexity to perform the "CAM-Convs", our method only requires to know the dataset baseline (distance between cameras). To train on multiple datasets, the only required step is to multiply the $P_a$ by the relative baseline with respect to a reference dataset. For instance, the baseline in the KITTI dataset is about 54cm, and, as mentioned before, we have set this baseline to correspond to $P_a^K = 153$. Then for the CityScapes dataset, whose baseline is 22cm, its pan amount will be given by $P_a^{CS} = (22/54)P_a^K$. For the VICLAB_STEREO dataset with the baseline of 12cm long, its pan amount becomes $P_a^{VL} = (12/54)P_a^K$. When training on KITTI + CityScapes (K+CS), a batch of size 8 contains 4 images from each dataset. When training on KITTI + CityScapes + VICLAB (K+CS+VL), each batch of size 8 contains 3, 3 and 2 images, respectively. For the comparison against the recent works, we train our model and the state-of-the-art models with KITTI (K) and KITTI + CITYSCAPES (K+CS) only, and evaluate the resulting trained models on the 400 images from the KITTI2015.

### A.4 IMPLEMENTATION DETAILS

For the training of all models, we used the Adam optimizer (Kingma & Ba, 2014) with the recommended $\beta$'s (0.5 and 0.999) for the regression task with a batch size of 8 for 50 epochs. The initial learning rate was set to 0.0001 and was halved at epochs 30 and 40. The following data augmentations on-the-fly were performed: Random resize with a factor between 0.5 and 1 conditioned by the subsequent 256x512 random crop; random horizontal flip, random gamma, and random color and RGB brightness. It was observed that vertical flip has made the learning more difficult, thus it was avoided. When training our model (the monster-net), the training images were sampled with a 50% chance for rightward ($P_a > 0$) or leftward ($P_a < 0$) panning. Additionally, random resolution degradation was applied to the input only by down-scaling followed by up-scaling back to its original size using a bicubic kernel with a scaling factor between 1 and 1/3 while keeping the target view unchanged. Random resolution degradation has improved our results by making the network more sensitive to edges and forcing it to focus more on structures and less on textures. Similar "tricks" have been used in previous works in the form of adding noise to the input of discriminators (Sønderby et al., 2017; Zhang et al., 2019) to make them invariant to noise, and more sensible to structures. When training our monster-net, either the left or the right view can be used as the input during training. When the left-view is used as the center input view $I_c$, the pan amount $P_a$ is set to 153 and the ground truth $I_{gt}$ is set to be the right image. In the opposite case, when the center view is the right image, $P_a$ is set to -153 and the left-view is set as the GT. For our monster-net, the "t-shaped" kernel was set to have short wings of 16 elements and a long wing of 32 elements plus one center element $T_c$. For the Deep3D, the 1D kernel is set to have 33 elements and 49 elements for the Deep3D-B variant. For the SepConv and the SepConv-D cases, we set the horizontal and vertical 1D kernels to have 1x51 and 51x1 shape, respectively, as in (Niklaus et al., 2017).

---

[1]https://www.mathworks.com/help/vision/ref/rectifystereoimages.html

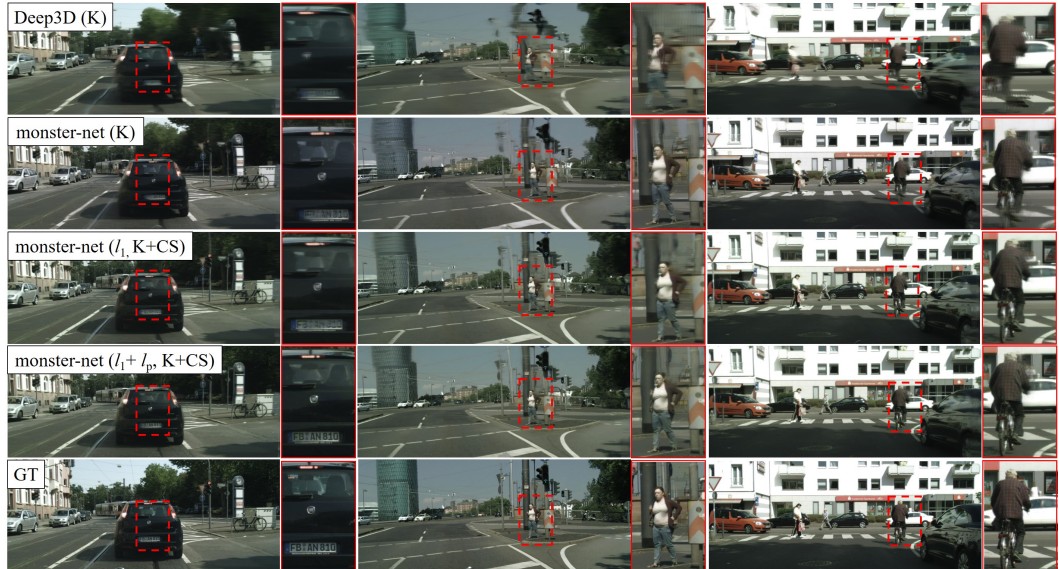

Figure 13: Results on the CityScapes dataset. Our method trained on KITTI-only (K), generalizes very well on the unseen images with an improvement over 3dB against the Deep3D baseline.

### A.4.1 LOSS FUNCTION

We train our monster-net with a combination of $l_1$ loss and perceptual loss (Johnson et al., 2016). The later measures the distance between the generated view ($\mathbf{I}_o$ or $\mathbf{I}_o^t$) and the ground truth ($\mathbf{I}_{gt}$) images in the deep feature space of a pre-trained network for image classification. The perceptual loss is especially good to penalize deformations, textures and lack of sharpness. The mean square error of the output of the first three max-pooling layers from the pre-trained $VGG19$ (Simonyan & Zisserman, 2014), denoted by $\phi^l$, was utilized as the perceptual loss function. To balance the contributions of the $l_1$ and perceptual losses, a constant $\alpha_p = 0.01$ was introduced. This combination of loss terms was applied to both the low-resolution panned image $\mathbf{I}_o^t$ and super-resolved panned image $\mathbf{I}_o$ to yield the total loss function $L_{pan}$ as follows:

$$L_{pan} = ||\mathbf{I}_{gt} - \mathbf{I}_o||_1 + ||\mathbf{I}_{gt}^{1/2} - \mathbf{I}_o^t||_1 + \alpha_p \sum_{l=1}^{3} ||\phi^l(\mathbf{I}_{gt}) - \phi^l(\mathbf{I}_o)||_2^2 + ||\phi^l(\mathbf{I}_{gt}^{1/2}) - \phi^l(\mathbf{I}_o^t)||_2^2 \quad (10)$$

where $\mathbf{I}_{gt}^{1/2}$ is the bilinearly downscaled version of the ground truth by a factor of 1/2.

### A.5 RESULTS ON THE CITYSCAPES DATASET

Visualizations on the CittyScapes (CS) datasets for our monster-net trained on KITTI (K) and on KITT + CityScapes (K+CS) are depicted in Figure 13. The First row of Figure 13 shows the synthesized images from the Deep3D baseline, it can be noted that it over-fits to the training baseline of the KITTI dataset, performing very poorly on CityScapes. The subsequent rows show the results for our monster-net when trained without and with the CityScapes dataset. Our models generate very good structures and sharp panned views as depicted the red highlighted regions on Figure 13 for both cases of with and without CityScapes training. When trained on KITTI-only, our method generalizes very well on the CityScapes dataset, with a performance improvement of 3dB over the Deep3D baseline as shown in Table 1. When trained on K+CS, we obtain an additional improvement of 4dB against the KITTY-only trained monster-net. Additionally, we present results for our monster-net trained with and without perceptual loss, ($L_1$) and ($l_1 + l_p$) respectively, on the CityScapes dataset. Sharper results with clear edges and structures are achieved when utilizing the perceptual loss, as depicted in the highlighted areas in Figure 13.

| GT | monster-net K+CS+VL | monster-net K+CS |
|:---:|:---:|:---:|
| (mean PSNR / SSIM) | (21.78dB / 0.718) | (19.92dB / 0.655) |

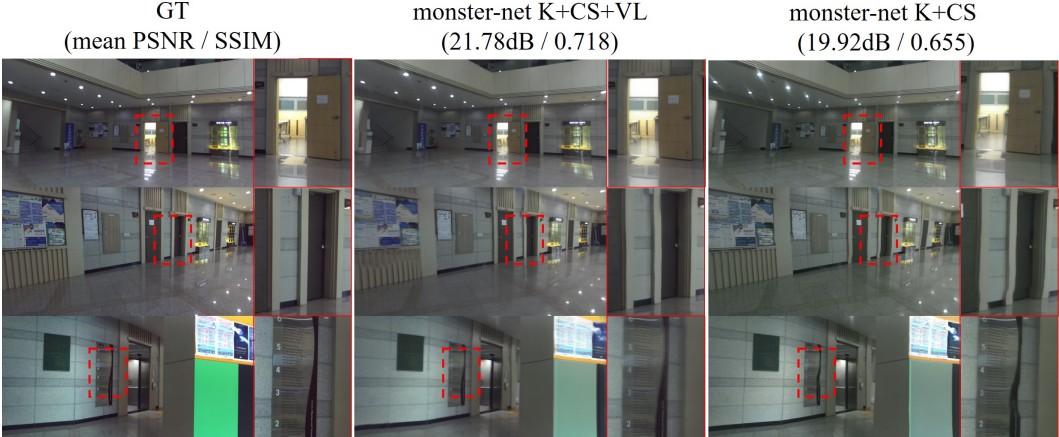

Figure 14: Results on the VICLAB_STEREO (VL) dataset. The monster-net trained on the K+CS+VL datasets achieves better structures in homogeneous areas (highlighted in red).

### A.6 RESULT ON THE VICLAB_STEREO INDOORS DATASET

A network that is trained on outdoor datasets only is not able to generalize well on highly homogeneous areas that are common in the indoors datasets but rare in the outdoor scenes. Visualization of the synthetic views generated for the VICLAB_STEREO (VL) indoors dataset is provided in Figure 14. We compare the results of our network trained on K+CS versus those of our monster-net trained on K+CS+VL. The latter achieves better generalization, as depicted in Figure 14, with marginal performance decrease in the KITTI dataset (-0.09dB) and considerable quality improvement on the VICLAB_STEREO dataset (+1.86dB), as showed in the last row of Table 1.

### A.7 3D PANNING BEYOND THE BASELINE

As our method allows for arbitrary camera panning, it is possible to perform 3D pan beyond the baseline as depicted in Figure 15, where the pan amount was set to go 30% beyond the baseline for both leftward and rightward camera panning, that is $P_a = -200$ and $P_a = 200$ for the per-scene top and bottom samples respectively, where the input image for both pan amounts was set to be the left-view. It is observed that the monster-net with adaptive dilations generates naturally looking new views with consistent structures and without discontinuities even at beyond training baseline 3D panning.

### A.8 ABLATION STUDIES

We demonstrate the contribution of our design choices in this section. Our main contribution is the adaptive "t-shaped" kernel equipped with globally and locally adaptive dilations. Figure 16-(a) shows the effect of adaptive dilations in comparison with the fixed dilation. As can be observed, the resulting synthesized image by a fixed local dilation kernel shows unnaturally looking regions with low contrast, discontinuities and blurs. Unlike any previous work, our pipeline can greatly benefit from training on multiple datasets at the same time, as shown in Figure 16-(b). That is, our method greatly benefits from training on two datasets (KITTI and CityScapes) as it exploits the baseline information via its global adaptive dilation property. Training on both KITTI and CityScapes contributes to improved geometry reconstruction as the network is exposed to a wide variety of objects at many different resolutions where, in addition to applying random resizing during training, the resolutions and baselines of these two datasets are very different. Figure 16-(c) shows the effect of utilizing our sr-block. Even if the quality of the panned image $\mathbf{I}_o^t$ is good in terms of structure, the sharpness is further improved by the addition of the super resolution block. Finally, we analyze the effect of the perceptual loss. By utilizing the perceptual loss, our monter-net is able to better synthesize rich textures and complex or thin structures, as depicted in Figure 16-(d), even though the PSNR and SSIM are slightly lower as shown in Table 1. The last is known as the "perception-distortion

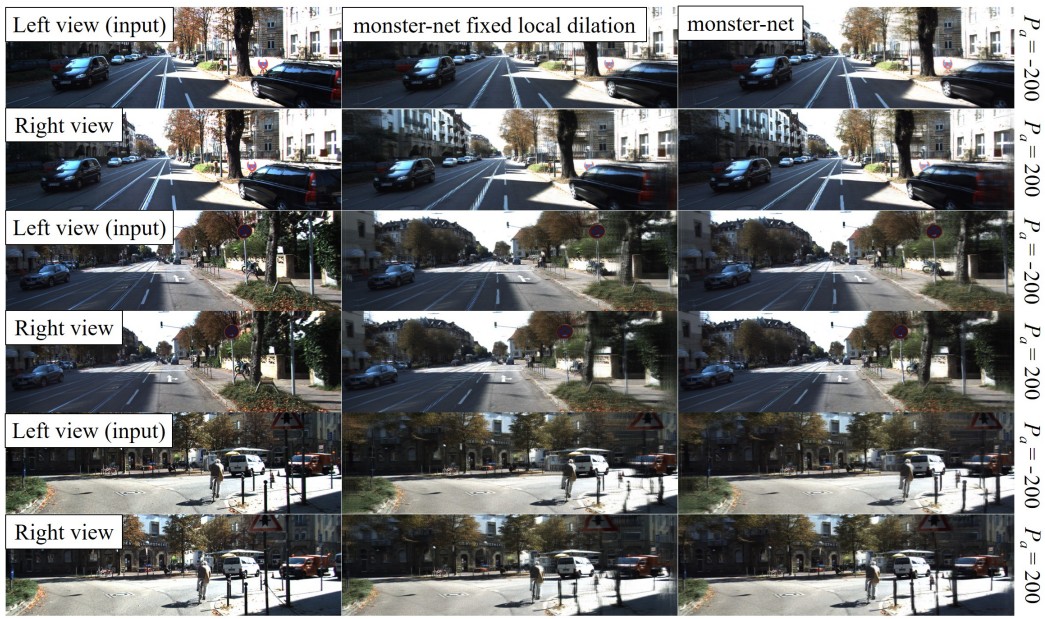

Figure 15: Our models generating 3D panned views at 30% beyond the baseline for the leftward and rightward camera panning. The magnification of the figures helps to better compare between the fixed local dilation monster-net and adaptive local dilation monster-net.

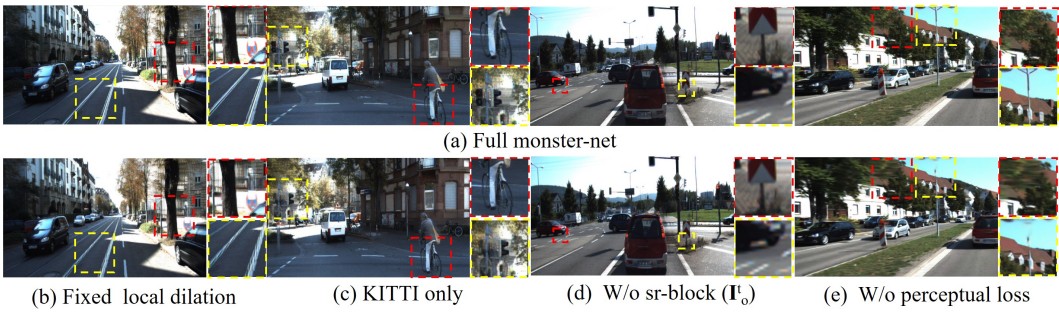

Figure 16: Ablation studies

tradeoff" (Blau & Michaeli, 2018) which suggests that better synthetic looking images not always yield higher PSNR/SSIM.

## A.9 DISPARITY/DEPTH ESTIMATION RESULTS

Our monster-net with refinement block beats the current state-of-the-art methods for unsupervised monocular depth estimation in terms of prediction accuracy for the KITTI2015 dataset. As shown in Table 3, the primitive disparity $D_p$, that can be extracted from the longer wing of the "t-shaped" kernel, is already among the best performing unsupervised methods. When we add the refinement block with ambiguity learning, our model results outperform those of the state-of-the-art. Furthermore, we get an additional improvement in the $a_1$ accuracy metric when we add our novel special-post-processing (spp) step. The qualitative comparison against previous methods and the ground truth disparity is shown in Figure 17. It is noted that our monster-net with disparity refinement and special-post-processing generates very reliable disparity maps even on thin structures and image borders. Additionally, our method benefits from very good detection of far away objects

Table 3: Disparity estimation performance on the KITTI2015 metrics from (Eigen et al., 2014). Models are trained with (V) video, (S) stereo, semi global matching (SMG) or GT depth (Supp), and can take stereo inputs (s), nine consecutive input frames during testing and training (9-view), or one input frame during testing and nine consecutive views as supervision during training (1/9-view). Additionally, (st) stands for student, (t) for teacher, (pp) for post-processing and (spp) for special post-processing. The best performing models in terms of $a^1$ threshold, which is the percentage of disparity values with a relative error less than 0.25, are highlighted in bold.

| Model | Supp | V | S | dataset | abs rel↓ | sq rel↓ | rms↓ | log rms↓ | $a^1$ ↑ | $a^2$ ↑ | $a^3$ ↑ |
|---|---|---|---|---|---|---|---|---|---|---|---|
| Pilzer et al. (2019) (st) | | | x | K | 0.142 | 1.231 | 5.785 | 0.239 | 0.795 | 0.924 | 0.968 |
| Ranjan et al. (2019) | | x | | K | 0.140 | 1.070 | 5.326 | 0.217 | 0.826 | 0.941 | 0.975 |
| Ranjan et al. (2019) | | x | | K+CS | 0.139 | 1.032 | 5.199 | 0.213 | 0.827 | 0.943 | 0.977 |
| Godard et al. (2017) | | | x | K | 0.149 | 2.565 | 6.645 | 0.245 | 0.849 | 0.936 | 0.969 |
| Godard et al. (2017) (pp) | | | x | K | 0.114 | 1.138 | 5.452 | 0.204 | 0.859 | 0.946 | 0.977 |
| Tosi et al. (2019) | SMG | | x | K | 0.111 | 0.867 | 4.714 | 0.199 | 0.864 | 0.954 | 0.979 |
| Gonzalez & Kim (2019b) | | | x | K | 0.113 | 1.114 | 5.364 | 0.195 | 0.866 | 0.951 | 0.981 |
| Godard et al. (2017) (pp) | | | x | K+CS | 0.100 | 0.934 | 5.141 | 0.178 | 0.878 | 0.961 | 0.986 |
| Wang et al. (2019a) (9-view) | | x | | K | 0.112 | 0.418 | 2.320 | 0.153 | 0.882 | 0.974 | 0.992 |
| Pilzer et al. (2019) (t) | | | x | K | 0.098 | 0.831 | 4.656 | 0.202 | 0.882 | 0.948 | 0.973 |
| Tosi et al. (2019) (pp) | SMG | | x | K+CS | 0.096 | 0.673 | 4.351 | 0.184 | 0.890 | 0.961 | 0.981 |
| ours w/o refine block | | | x | K+CS | 0.121 | 1.028 | 4.917 | 0.174 | 0.885 | 0.969 | 0.989 |
| ours with refine block | | | x | K+CS | 0.098 | 0.893 | 4.836 | 0.166 | 0.894 | 0.967 | 0.988 |
| ours with refine block (pp) | | | x | K+CS | 0.095 | 0.793 | 4.634 | 0.159 | 0.896 | 0.969 | 0.989 |
| **ours with refine block (spp)** | | | x | K+CS | 0.099 | 0.950 | 4.739 | 0.160 | **0.900** | 0.971 | 0.989 |
| Gur & Wolf (2019) | x | | | K | 0.110 | 0.666 | 4.186 | 0.168 | 0.880 | 0.966 | 0.988 |
| Luo et al. (2018) | x | | | K | 0.094 | 0.626 | 4.252 | 0.177 | 0.891 | 0.965 | 0.984 |
| **Wang et al. (2019a) (1/9-view)** | x | x | | K | 0.088 | 0.245 | 1.949 | 0.127 | **0.915** | 0.984 | 0.996 |
| Godard et al. (2017) (s) | | | x | K | 0.068 | 0.835 | 4.392 | 0.146 | 0.942 | 0.978 | 0.989 |
| Lai et al. (2019) (s) | | x | x | K | 0.062 | 0.747 | 4.113 | 0.146 | 0.948 | 0.979 | 0.990 |
| **Wang et al. (2019b) (s)** | | x | x | K | 0.049 | 0.515 | 3.404 | 0.121 | **0.965** | 0.984 | 0.992 |

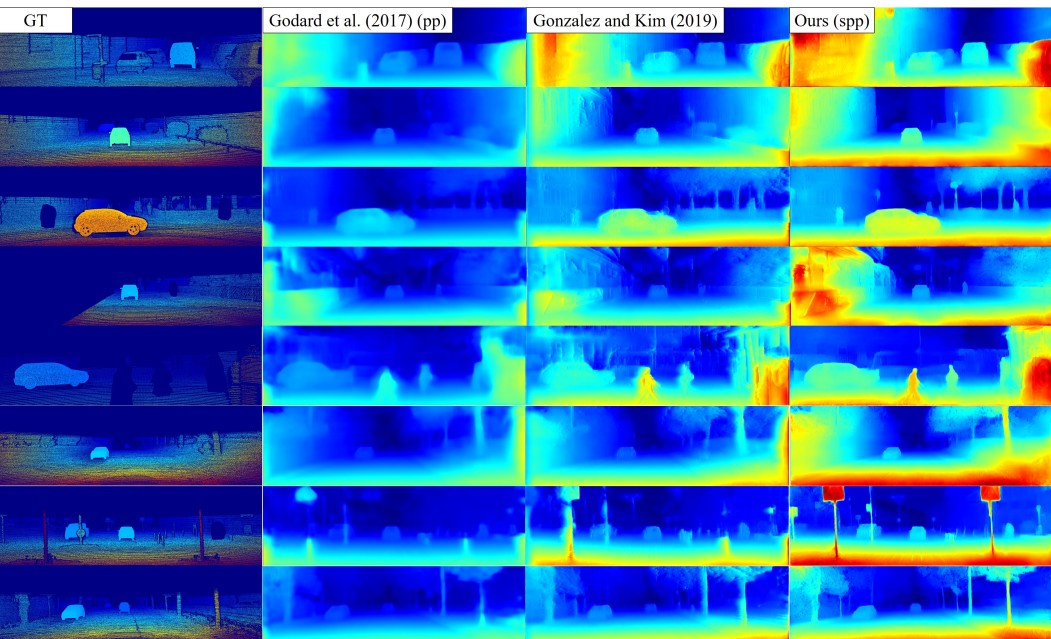

Figure 17: Qualitative comparison between our method and the SOTA for the unsupervised monocular depth estimation task on the KITTI2015 dataset.

