# OpenReview forum: "Deep 3D Pan via local adaptive "t-shaped" convolutions with global and local adaptive dilations"
_ICLR.cc/2020/Conference — Accept (Poster)_

### Official Review · AnonReviewer1 · 2019-10-22
**Official Blind Review #1**

**Rating:** 6

**Review:**

The paper considers the problem of performing stereoscopic view synthesis (i.e., generating a new view seen from a different camera position) at an arbitrary position along the X-axis from a single input image only. This is an important problem as it enables 3D visualization of a 2D input scene. The paper focuses on the particular problem of generating a stereoscopic view from a single image (i.e., a right and left view from a center image).
For this purpose, the paper proposes a t-net architecture which is an autoencoder or U-net like architecture that estimates the values for the t-convolutions proposed in the paper. The network (called monster-net) takes a center image and a pan amount as input, and from those synthesizes the image with the respective view.

The paper demonstrates that their idea of t-convolutions outperforms recent competing approaches such as deep 3D on available datasets as well as on an in-house collected dataset. The figures provided demonstrate that the views generated by the proposed Monster-net visibly look slightly better than those generated by the competing approaches DeepD and SepConv. In addition, the paper is well written and easy to follows. I therefore recommend acceptance of this paper. I would like to emphasize that while I work in deep learning, I don't work on view synthesis and therefore it is difficult for me to evaluate the novelty of the proposed approach as well as the difficulty of the problem.

**Experience Assessment:**

I do not know much about this area.

**Review Assessment: Checking Correctness Of Derivations And Theory:**

N/A

**Review Assessment: Checking Correctness Of Experiments:**

I assessed the sensibility of the experiments.

**Review Assessment: Thoroughness In Paper Reading:**

N/A

---

### Official Review · AnonReviewer2 · 2019-10-23
**Official Blind Review #2**

**Rating:** 6

**Review:**

The submission proposes a method to perform stereoscopic view synthesis. The method consists of a neural network model that estimates a novel viewpoint either to the left or to the right of a given image. The two key insights of the proposed method is 1) to learn the weights of a t-shaped kernel when performing novel view synthesis, and 2) to estimate and use adaptive dilations on those kernels.

The proposed approach is sound and the evaluation methodology used is adequate. The technique proposed is somewhat related to the recent interest in the community to apply CNNs to non-regular grids [1,2,3].

Sec. 4.2 states that the proposed method outperforms other existing methods on monocular depth estimation, while the table seems to indicate that other methods (e.g., Wang et al. 2019) obtain a better a1 measure.

I would refrain from calling a 0.004 increase in a1 “a remarkable improvement” (sec. A.9).

Minor details
- I would refrain from using the word “prove” (abstract, sec. 4), since no proof is provided. “demonstrate”
- p. 2 “is open known to be a much more complex problem”: I think the authors meant either “is known to be a much more complex problem” or “is still an open problem”?
- p. 9 “planed” should be written “planned”.
- p. 9 “the receptive field [...] is of the 153x153 size”: should be “has a size of 153x153”.


[1] Su, Yu-Chuan, and Kristen Grauman. "Learning spherical convolution for fast features from 360 imagery." Advances in Neural Information Processing Systems. 2017.
[2] Coors, Benjamin, Alexandru Paul Condurache, and Andreas Geiger. "Spherenet: Learning spherical representations for detection and classification in omnidirectional images." Proceedings of the European Conference on Computer Vision (ECCV). 2018.
[3] Zhao, Qiang, et al. "Distortion-aware CNNs for Spherical Images." IJCAI. 2018.


**Experience Assessment:**

I have read many papers in this area.

**Review Assessment: Checking Correctness Of Derivations And Theory:**

I assessed the sensibility of the derivations and theory.

**Review Assessment: Checking Correctness Of Experiments:**

I assessed the sensibility of the experiments.

**Review Assessment: Thoroughness In Paper Reading:**

I read the paper thoroughly.

---

> ### Author Response · Authors · 2019-11-09
> **Reply to reviewer #2**
>
> Thank you for your comments.
> It seems that there is a misunderstanding regarding the values of Table 2. The method of Wang et al 2019, is a supervised method, that is, they train with the depth ground truth. Our depth estimation is unsupervised and belongs to a different category as signalized by the division lines in the table. Our method is compared to Tosi et al. 2019 as theirs is also unsupervised. It is worth to mention that Wang et al 2019 not only trained with depth ground truth but also with 9 consecutive views for additional supervision, thus exploiting temporal consistencies. Our method, with no ground truth and only 2 views during training, performs remarkably well.
>
> Regarding the comment in section A.9, our novel post-processing step remarkably improves over the naive post-processing used in previous works as visualized in Figure 12, by removing most depth “shadows”. However, due to the sparsity of the evaluation ground truth, the metric numbers seem to only slightly improve, as you noted.
>
> Regarding the minor details, we will use the correct wording, instead of “prove” we will change to “demonstrate”. The same goes for “is known to be a much more complex problem”, “planned”, and “has a size of 153x153”.

---

### Official Review · AnonReviewer3 · 2019-10-24
**Official Blind Review #3**

**Rating:** 3

**Review:**

Summary:
This paper proposes a deep learning method to produce "pans" of an input image. That is, simulated images of the scene from translated viewpoints. Unlike some previous work that considers only a fixed baseline (such as the 2nd view of a stereo camera), this approach allows generation of a range of views. A specially crafted convolutional architecture is shown to be well-suited to this problem. Results demonstrate visually pleasing image generation and low metric errors on several datasets.

Strengths:
- The justifications for the design choices in this paper, in particular the convolution structure and connection to image geometry, was quite strong compared to recent papers (although, many of the presented ideas are known in more classical, non-learning, techniques).
- All presented empirical results are impressive, and show the method is likely to "really work" and be reproducible, judging from the number of experiments where the method has consistently outperformed.
- The method is clear and straightforward to implement either on its own, or as a module/architecture within a larger pipeline.

Areas for Improvement and Detailed Suggestions:
- The problem of panned view generation is a bit more narrow than some authors are lately considering (generate any viewpoint including off-axis rotations).
- The t-shaped network architecture here is largely presented as only appropriate to handing image panning. Could a more general network be created, perhaps parameterized by the type of rigid motion occurring? Better, could more flexible networks be proposed with sparsity constraints that allow sub-patterns like the t-network to be learned in a data-driven fashion?
- Please try to cite referenced work more consistently. For several methods, such as Deep 3D Zoom Net, you have begun to discuss the method using name only for large stretches. It would be helpful to keep using the citation at least once per paragraph to remind the reader of the source of these ideas.

Decision and Justification:
Weak reject due to the lack of generality in the approach. I suspect the impact of this work will be a bit less than the very competitive bar for ICLR this year. However, I must note that I am the least expert in this area, out of any paper in my stack.



**Experience Assessment:**

I have read many papers in this area.

**Review Assessment: Checking Correctness Of Derivations And Theory:**

I assessed the sensibility of the derivations and theory.

**Review Assessment: Checking Correctness Of Experiments:**

I carefully checked the experiments.

**Review Assessment: Thoroughness In Paper Reading:**

I read the paper at least twice and used my best judgement in assessing the paper.

---

> ### Author Response · Authors · 2019-11-09
> **Reply to reviewer #3**
>
> Thank you for your kind review.
> Our “t-shaped” adaptive kernel equipped with adaptive dilations can be trivially generalized to any camera translation (Y and Z axis) by simply allowing the kernel to rotate. E.g. when moving the camera in the Y-axis the long wing of the t-shaped kernel would point towards the Y-axis, and when moving the camera in the Z-axis the t-shaped kernels would point to the center of the image. In this work, we concentrated on horizontal stereoscopic view synthesis as it finds more direct applications in the real world.
>
> The impact of our work goes beyond image synthesis, as we also present better than state-of-the-art results for monocular depth estimation when no ground truth depth is available (unsupervised approach) and present a way to learn view synthesis and depth from many datasets with different camera baselines simultaneously.
>
> To put our paper in perspective with the most recent works, Choi et al. proposed “Extreme view synthesis” where, given two images with a narrow baseline as input (1.4cm), they generate a 30x extrapolated view, that is a new view with a baseline of 42cm. Our work, from a single image, generates a novel view with a baseline of 54cm (KITTI baseline), and even at 30% beyond the baseline (~70cm) our method still generates a decent image. Unfortunately, as [1] was made public after our ICLR submission, it is not yet included in our related works section..
>
> Finally, regarding your comment about the citations, we will increase the citation frequency in the final version of our paper.
>
> [1] Choi, Inchang, et al. "Extreme view synthesis." Proceedings of the IEEE International Conference on Computer Vision. 2019.

---

### Author Response · Authors · 2019-11-15
**Revision uploaded on the 12th**

Please take a quick look at the revision uploaded on the 12th where we included the comments from:
* Reviewer 3 on extending our t-shaped kernel to any rigid motion and increased citation frequency (we are a little bit limited here as we run out of space due to the ICLR citation style).
* Reviewer 2 on minor writing errors.

 Additionally, we added more visualizations of the predicted depth maps in Figure 17.

---

### Decision · Program_Chairs · 2019-12-19

**Decision:**

Accept (Poster)

**Comment:**

Two reviewers recommend acceptance while one is negative. The authors propose t-shaped kernels for view synthesis, focusing on stereo images. AC finds the problem and method interesting and the results to be sufficiently convincing to warrant acceptance.